# Chromosomal Instability and Telomere Attrition in Systemic Sclerosis: A Historical Perspective

**DOI:** 10.3390/genes16121466

**Published:** 2025-12-08

**Authors:** Carol M. Artlett

**Affiliations:** Department of Microbiology and Immunology, Drexel University, Philadelphia, PA 19129, USA; cma49@drexel.edu

**Keywords:** chromosomal instability, telomere attrition, systemic sclerosis, fibrosis, SSc autoantibodies

## Abstract

Background/Objectives: Systemic sclerosis (SSc) is a rare, complex autoimmune disease characterized by fibrosis of the skin and internal organs. While its pathogenesis is not fully understood, chromosomal instability and telomere attrition have emerged as significant areas of investigation. Methods: This review provides a historical narrative perspective and synthesizes current findings on the role of these genomic anomalies in SSc pathogenesis. We synthesized findings from foundational and recent research articles investigating genotoxic factors, chromosomal aberrations, and telomere biology in SSc. Results: There is a strong historical basis for chromosomal instability in SSc, manifesting as micronuclei, translocations, and breaks. This instability is driven by clastogenic factors and oxidative stress. SSc-specific autoantibodies are implicated; anti-centromere antibodies correlate with aneuploidy and micronuclei, while anti-topoisomerase I may inhibit DNA repair. SSc is also characterized by significant telomere attrition, first reported in 1996 and now confirmed by additional genetic studies. This telomere loss is associated with reduced telomerase activity and the presence of autoantibodies against telomere-associated proteins, including shelterin components. Conclusions: We conclude that inflammation, telomere attrition, and chromosomal instability are linked in a self-perpetuating cycle that drives SSc pathogenesis. We propose that an initial inflammatory stimulus leads to reactive oxygen species production, causing telomere damage and attrition. Critically short telomeres trigger faulty DNA repair mechanisms, such as breakage–fusion–bridge cycles, resulting in chromosomal instability. This genomic damage, in turn, acts as a danger signal, further activating inflammatory pathways and creating a feedback loop that perpetuates fibrosis.

## 1. Introduction

Systemic sclerosis (SSc) is a rare autoimmune disease that causes fibrosis in the skin and internal organs. Specific autoantibodies develop, which frequently predict the SSc subtype [1]. Raynaud’s phenomenon is one of the earliest symptoms, often occurring before or concurrently with skin thickening [2]. Other features of SSc, such as esophageal dysmotility, kidney failure, and pulmonary and cardiac fibrosis, are associated with this disease and have garnered considerable attention over the decades, likely because they significantly affect the morbidity of the patient [3]. SSc has two major subsets: limited cutaneous (lcSSc) and diffuse cutaneous (dcSSc) disease. In lcSSc, the skin involvement is primarily contained to the face and below the knees and elbows [4]. In this disease subset, organ involvement develops slowly, and the most common autoantibody targets centromeric proteins. In contrast, dcSSc is a rapidly progressing phenotype characterized by widespread skin and internal organ involvement [4]. This subset is primarily associated with autoantibodies that target topoisomerase I and RNA polymerase III.

SSc has been extensively characterized, but more recently, chromosomal instability and telomere attrition have come to the forefront, prompting investigators to examine these anomalies in SSc. This review synthesizes and discusses the findings of these chromosomal aberrations and telomere attrition in SSc, providing a historical narrative of their investigation and current findings.

## 2. Chromosomal Abnormalities in SSc

The most frequently observed chromosomal rearrangement in the healthy population is balanced translocations and an inversion of chromosome 9. Balanced translocations include the Robertsonian translocation between chromosomes 13 and 14 (rob(13;14)(q10;q10)) [5]. It is found in approximately 1 in 800 individuals [6]. Another translocation occurs between chromosomes 11 and 22, designated as t(11;22)(q23;q11), which is more frequently found in males [7]. The chromosome 9 inversion inv(9)(p11q12) [5] is found in about 1.5% of the general population [8]. However, although these chromosomal rearrangements are present in the general population, the frequency of random chromosomal abnormalities can vary with age. For example, chromosomal aberrations have been reported to be three times higher in older individuals (mean age 75 years) than in younger individuals (mean age 28.5 years [9]. The onset of SSc typically occurs later in life, between the ages of 40 and 60 [10]. When it happens in the elderly (>60 years), it is associated with accelerated disease progression [10,11]. SSc does not have a familial inheritance pattern, and although rare, there are reports of families with more than one affected individual [12,13]. No single, disease-defining chromosomal rearrangement is consistently found in patients.

The Emerit group first studied chromosomal abnormalities in SSc [14,15,16,17]. They identified acentric fragments, dots, dicentric, and ring chromosomes. Other abnormalities seen were the translocation or deletion of chromosomal fragments and intrachromosomal rearrangements [16]. A microchromosome derived from chromosome 11 was reported in a patient [18]. This microchromosome was devoid of telomeric sequences and was somatically stable [18]. The telomeric fusions between sister chromatids in SSc cells have been shown to last for more than two cell generations [19]. Micronuclei are frequently seen in both lcSSc and dcSSc cohorts [20,21,22]. Overall, the chromosomal abnormalities were not confined to lymphocytes but were also found in cultured and uncultured fibroblasts and bone marrow [17]. Micronuclei are found in equal numbers in affected and unaffected fibroblasts [23]. The frequency of abnormalities in various SSc cohorts was significant, and in one instance, it was found to be as high as 95% [24]. First-degree family members also exhibited increased chromosomal aberrations, with 86% found in siblings and 68% in offspring [17], suggesting an environmental or familial abnormality.

Historically (Figure 1), the study of chromosomal instability in SSc started in 1971, when Emerit identified a clastogen in the sera and cell extracts from SSc patients [16]. The clastogenic factor in the sera was later determined to be inosine triphosphate and inosine diphosphate. When added to cell cultures from healthy donors, these compounds induced chromosomal breakage in a dose-dependent manner [25]. Emerit was also the first to propose that reactive oxygen species (ROS) were responsible for sister chromatid exchange [23] and could be a key driver of the genotoxic effects observed in SSc [26]. The discovery that superoxide dismutase could reduce chromosomal aberrations was significant, highlighting ROS as a key driver [27]. The oxidative metabolic activity found in the blood of SSc patients was significantly higher than that of normal individuals [28], and these patients produce more of the superoxide anion (O_2_ *-) [29] and have more active NADPH oxidase activation [30,31]. Another study found that patients had lower overall levels of superoxide dismutase activity and total antioxidant activity [32], suggesting that ROS overwhelmed the patients’ antioxidant defenses, thereby increasing their risk of additional genotoxic effects. In SSc, urinary levels of 8-hydroxy-2′-deoxyguanosine, a marker of oxidative DNA damage, were significantly higher than those in controls [33].

Several pre-clinical studies have investigated various antioxidants as potential therapeutics for SSc. The antioxidant N-acetylcysteine supported the findings that this molecule reduced fibrosis in both in vitro and in vivo studies [34]. Epigallocatechin-3-gallate, a potent antioxidant and the most abundant polyphenol in green tea, was found to reduce collagen expression, collagen gel contraction, and suppress intracellular ROS, ERK1/2 kinase signaling, and NF-κB activity in SSc cells [35]. One open-labeled study demonstrated an improvement in skin thickness in SSc patients who applied a cream containing 0.6 mg/mL superoxide dismutase [36]. The caveats to this study were that it was open-label and involved a small patient cohort. Although intriguing, to date, no additional studies using this cream have been undertaken to validate this observation. Overall, the studies suggest that redox regulation plays a strong role in the pathogenesis of SSc and may contribute to the development of chromosomal abnormalities.

Using an assay to measure acquired genetic damage, Roberts-Thomson et al. [37] found that SSc patients had a higher proportion of mitotic mutations in the glycophorin A gene. This mutation rate was higher than that of the age-matched controls, and the authors propose an aberrant response to DNA damage and repair in this disease. More recent analyses have shown that SSc has genomic alterations in the variable tandem number repeats [38]. All members in each family analyzed were confirmed using HLA typing. This study was the first to demonstrate that the tandem repeats could be altered in approximately 40% of SSc patients and fibroblast cell lines, with lymphocytes also being affected. Some of the repeats were correctly transmitted through the family, but had an altered size in the proband [38]. In a study using the comet assay, Palomino et al. [39] reported that SSc patients had increased DNA damage, and it was equally prevalent in both lcSSc and dcSSc. The authors proposed that patients with SSc had a less efficient DNA repair process, thus leading to the increased genomic instability seen in lymphocytes [39]. Hypermutation is substantially increased in SSc-lesional skin fibroblasts, and these mutations occur in genes involved in genomic integrity (histone integrity, histone acetylation, helicase, and DNA methylation), comparable to levels observed in some cancers [39].

Loss of centromeric sequences was recently demonstrated in dcSSc patients, with significant deletions on chromosomes 1 and 2, and a specific centromeric expansion on chromosome 19 was also noted [22]. Aneuploidy was reported in 10–50% of chromosome spreads in dcSSc, and there was a loss of centromere identity in the observed micronuclei. The extent of the micronuclei positively correlated with ROS levels [22]. The most recent study in 2024 reported an increase in single-base substitutions (SBS) in SSc patients [40]. The most prevalent signature was the SBS93 mutation signature, which is a prominent feature found in various cancers (Catalog of Somatic Mutations in Cancer, Wellcome Sanger Institute, Saffron Walden, UK). This study also found that doublet base substitutions, insertions, and deletions were elevated in SSc, along with mutation clustering. Loss of heterozygosity was also more often seen in SSc fibroblasts [40]. Overall, like the authors, we also question whether chromosomal instability also places the patient at risk for cancer, as the incidence of cancer is markedly increased in SSc patients [41,42,43,44].

Overall, there is likely an unknown environmental factor, alongside other significant risk factors, such as genetics, which contribute to chronic inflammation in SSc. Chronic inflammation enhances fibroblast proliferation and differentiation and increases the release of ROS and clastogenic factors that damage DNA. The damaged DNA leads to chromosomal abnormalities seen in SSc cells, including acentric fragments, dots, dicentric and ring chromosomes, and increased sister chromatid exchange (Figure 2).

## 3. Key SSc Autoantibodies and Chromosomal Instability

Two of the most frequently observed autoantibodies in SSc patients are those targeting centromeric proteins and topoisomerase I. These autoantibodies tend to be mutually exclusive; however, when found together, patients exhibit a more severe form of the disease [45]. An additional antibody, less frequently observed, is directed at RNA polymerase III [46].

Autoantibodies were previously considered impervious to the nucleus; however, evidence suggests otherwise. A subset of anti-DNA autoantibodies in systemic lupus erythematosus was found to penetrate the nuclei of live cells, damaging the DNA or inhibiting its repair [47,48,49,50,51,52]. This suggests that the autoantibodies found in SSc may also penetrate the nucleus and affect their target antigens; however, this has not yet been proven for all SSc autoantibodies.

### 3.1. Centromeric Autoantibodies

The autoantibodies (CENP-A, CENP-B, and CENP-C) against centromeric proteins are primarily found in the lcSSc subset and have been shown to correlate with aneuploidy and chromosomal breaks in SSc patients [53]. Patients who were positive for CENP-A, CENP-B, or CENP-C had significantly more aneuploidy than those who were negative for these autoantibodies (e.g., the dcSSc cohort). However, SSc patients overall (those positive for centromeric autoantibodies and those negative) had substantially greater aneuploidy than the control group. The authors propose that a correlation exists between the presence of autoantibodies to CENP-A, CENP-B, or CENP-C and chromosomal aneuploidy, suggesting that aneuploidy may result from nondisjunction secondary to centromeric dysfunction [53]. In support of this hypothesis, they found that patients with an autoantibody to CENP-C had a higher incidence of chromosomal aneuploidy compared to those without this autoantibody. CENP-A and CENP-B are present on both functional and inactivated centromeres. In contrast, CENP-C localizes to the inner plate of the kinetochore of only functional centromeres, suggesting a causal link to aneuploidy [53]. Further studies have now shown that CENP-A also localizes to the inner plate of the kinetochore at the centromere, which is associated with a satellite DNA complex [54]. In contrast, CENP-B localizes to centromeric heterochromatin beneath the kinetochore [55].

Micronuclei are predominantly found in SSc cells and are thought to reflect the chromosomal damage observed in this disease. Those with lcSSc had significantly higher micronuclei frequencies than those with dcSSc. These results appeared to reflect the autoantibody profile, in which ACA-positive SSc patients had higher micronuclei frequencies than ACA-negative patients [21]. These results support the authors’ proposal that micronuclei correlate with the autoantibody profile and likely cause cytogenetic damage. These observations were confirmed in two additional studies by Fagone [20] and Patterson [56], where both reported a high correlation with micronuclei and the centromeric autoantibody. These key observations are presented in Table 1.

### 3.2. Topoisomerase I Autoantibodies

Topoisomerase I catalyzes the nicking of DNA, promoting its relaxation and unwinding. This relieves torsional strain in supercoiled DNA during transcription and replication [57]. The autoantibody was initially designated as Scl70, as it was identified in the sera of dcSSc patients and recognized an antigen that had a molecular weight of approximately 70 kDa [58]. This antibody was later identified to be topoisomerase I [59]. Recent studies by May et al. found that Scl70 can penetrate the nucleus of cells, directly inhibiting the activity of topoisomerase I [60]. These elegant studies showed that the transfer of the autoantibody into the nucleus, at least in part, seemed to be dependent on lipid rafts [60]. Currently, there are no direct assessments of chromosomal abnormalities associated with this autoantibody, and, to date, only associations have been reported. In patients with the topoisomerase I autoantibody, there was a prevalence of unstable DNA breaks, overall supporting a clastogenic effect on DNA and the possible interference with protective cellular mechanisms that typically stabilize DNA breaks [20]. Micronuclei have been reported in dcSSc cohorts with the Scl70 autoantibody, but at a lower frequency than in those with the centromeric autoantibody [20,21,56].

However, we speculate that if Scl70 can enter the nucleus and inhibit topoisomerase I activity, it may reflect some of the DNA-damaging effects observed with topoisomerase I inhibitors used in cancer treatments. Topoisomerase I causes single-strand nicks to relax the DNA for replication [61]. Chemical inhibitors of topoisomerase trap the enzyme after it has made the DNA cut, preventing it from resealing the DNA. This creates the Topo I-DNA cleavable complex [62,63]. When the DNA replication fork encounters this complex, it collapses, converting the initial single-strand nick into a double-strand break [62]. The consequences of this process lead to chromosomal fragmentation, translocations, and sister chromatid exchanges [64]. A recent study showed that the topoisomerase autoantibody prevents the formation of the Topo I-DNA cleavable complex [60]. When the replication fork encounters transcription complexes, this causes genomic instability [65]. A summary of these key observations is presented in Table 1.

**Table 1 genes-16-01466-t001:** Key observations between chromosomal instability and the SSc autoantibody profile.

Autoantibody Profile	Key Findings and Implications	References
Autoantibodies against centromeric proteins	CENP-A, CENP-B, and CENP-C autoantibodies can penetrate the nucleus and are associated with increased aneuploidy, chromosomal breaks, and elevated micronuclei	[20,21,53,56]
Autoantibody against topoisomerase I (Scl70)	Scl70 can penetrate the nucleus. Scl70 can prevent the formation of the Topo I-DNA complex, impede replication, and cause genomic instability. Patients with Scl70 have been associated with unstable DNA breaks and micronuclei.	[60]
RNA polymerase III	RNA polymerase III can repair double-stranded DNA breaks. It is currently unknown whether this autoantibody penetrates the nucleus, but if it does, it could affect RNA polymerase III function, contributing to DNA damage.	[66]

### 3.3. RNA Polymerase III Autoantibodies

The presence of an autoantibody that targets RNA polymerase III is associated with a severe and rapidly progressive form of SSc, characterized by widespread and rapid skin thickening and renal crisis, which is marked by acute kidney failure and malignant hypertension [67,68]. This autoantibody is found in approximately 11% of patients [69], but its prevalence varies by ethnicity and geographic location [70]. RNA polymerase III is a housekeeping gene that specializes in transcribing 5S rRNA, transfer RNAs, and the U6 spliceosomal RNA, all of which play a primary role in translation and related biological processes [71]. Currently, no direct studies have shown that this autoantibody penetrates the nucleus; however, if it did, it would likely cause substantial disruption to DNA repair, as emerging research has demonstrated that RNA polymerase III also plays a role in repairing DNA double-strand breaks [66]. Thus, if the RNA polymerase III autoantibody could cross the nuclear membrane, it could compromise the cell’s ability to repair DNA damage, leading to genomic instability. A summary of these key observations is presented in Table 1.

## 4. SSc and Telomere Attrition

Telomeres were first discovered in the 1930s by Hermann Müller [72]. They are repetitive hexameric DNA units located at the ends of chromosomes, serving to protect chromosome ends from being recognized and improperly repaired as double-stranded breaks. The most well-known of these sequences is the repeat sequence TTAGGG, which extends between 10 and 15 kilobases in newborns [73]. It is this sequence that is primarily lost during DNA replication. However, the telomere cap also comprises other non-randomly distributed consensus sequences. These are found in the subtelomeric regions and contain TTGGGG and TGAGGG repeat sequences, which play crucial and nuanced roles in regulating the structure and function of the chromosome ends. The telomere is stabilized by shelterin, and the shift from high-affinity (TTAGGG) to low-affinity variants (TTGGGG and TGAGGG) helps mark the transition from the telomere to the rest of the chromosome. Furthermore, reduced shelterin binding in this subtelomeric region can make the chromatin more open and accessible, allowing other regulatory proteins to bind. The G-rich strand overhangs the C-rich strand by 12–16 residues [74] and plays a central role in the function of the telomere [75]. With every round of cell division, 50–200 nucleotides of the TTAGGG sequence are lost. This shortening acts as a biological clock, and once telomeres become critically short, the cell’s DNA damage response is activated, triggering either cellular senescence or apoptosis [76]. Thus, telomere length predicts a cell’s replicative capacity [77].

To shield the telomeres from this response, a protein complex called shelterin binds to the chromosome ends. In some cells, the enzyme telomerase, a reverse transcriptase that uses an RNA template to synthesize new telomere DNA, counteracts this shortening. Telomerase is highly expressed in embryonic stem cells and specific adult cells with high turnover, such as lymphocytes. In most somatic cells with low telomerase activity, the alternative lengthening of telomeres pathway can maintain telomere length via homologous recombination. However, before cell death or senescence occurs, the shortened telomeres can lead to chromosomal aberrations, such as dicentric chromosomes, chromosomal breaks, deletions, and translocations [78,79,80].

Telomerase is an enzyme that elongates telomeres. In adults, it is highly active when cellular longevity is needed, such as during high rates of division, as in stem cells and germ cells. Telomerase contains a telomere-specific reverse transcriptase that adds repeats to the end of the telomeres. Telomerase activity is regulated by the shelterin complex, which comprises six proteins. These proteins are telomeric repeat binding factor-1, telomeric repeat binding factor-2, protection of telomeres-1, adrenocortical dysplasia homolog, TERF1-interacting nuclear factor-2, and repressor/activator protein-1 [81].

SSc was first associated with shortened telomeres in 1996 [82]. At the time, we hypothesized that the telomere attrition we observed could be the source of chromosomal damage and various aberrations resulting from telomeric fusions [82]. We found that the average loss of telomeric sequences was approximately 3 kb greater than in controls, and this was not related to patient age or disease duration [82]. Intriguingly, when analyzing peripheral blood mononuclear cells by Southern blotting, we did not observe a general shift in the telomeric smear, as seen in tumor cell lines [72], but instead we saw a broadening of the smear. We speculated at the time that this could imply that not all chromosomes have shortened telomeres, or not all cells have undergone telomere attrition. We also found that SSc family members, including spouses, had shorter telomeres than controls [82], suggesting an environmental exposure related to the familial chromosomal abnormalities identified by Emerit [17].

One investigation found that a subset of SSc patients had shorter telomeres in lymphocytes but not granulocytes [83]. This observation is interesting because granulocytes, particularly neutrophils, have a significantly faster, more dramatic turnover rate during inflammation than lymphocytes. Neutrophils may play a role in SSc, and they are found to be elevated, correlating with disease severity and tissue damage [84]. As they are elevated and turn over more rapidly, it would be expected that this cell phenotype exhibits telomere attrition.

Two more studies found shorter telomeres in SSc patients with interstitial lung disease and reported them to be substantially shorter in lcSSc than in dcSSc, even when age was taken into consideration [85,86]. Surprisingly, telomere attrition in the Liu study [85] study was greater in patients who did not exhibit any of the common autoantibody profiles associated with SSc. Furthermore, using longitudinal analysis, they found that shorter telomeres were associated with an increased risk for worsening interstitial lung disease [85]. Further support for telomere loss in this study reports substantial loss of telomeric sequences in both lcSSc and dcSSc [87]. Like the first publication investigating telomere loss in SSc [82], they also did not find a correlation between telomere attrition and disease duration [87]. However, contrary to these studies, MacIntyre et al. [88] found increased telomere lengths in an lcSSc cohort, and it is the only research to report this finding. To overcome these discrepancies, a Mendelian randomization study was conducted in 2023 to investigate the relationship between telomere length and SSc [89]. This study reported that shorter telomere lengths were associated with an increased risk of SSc onset, overall supporting the generalized finding that shorter telomeres are found in SSc [82,83,85,87]. However, contrary to the MacIntyre study [88], they observed that the lcSSc patients had shorter telomeres than controls [89]. In a study investigating different connective tissue diseases (rheumatoid arthritis, systemic lupus erythematosus, Sjogren’s syndrome, and SSc), telomerase activity in SSc was found to be significantly lower in lymphocytes than in the other connective tissue diseases and was substantially lower than that of the control cohort [90] and this may be associated with genotype frequencies in the telomerase gene [91].

### Autoantibodies Targeting Telomere-Associated Proteins in SSc

Recently, researchers have identified a range of autoantibodies targeting telomerase and its associated proteins, shedding new light on the pathogenesis of SSc and its association with shortened telomeres [92,93,94], and suggesting a potential role for telomere dysfunction in the underlying disease processes. Although rare, these autoantibodies target key components of the cellular machinery that maintain telomeres. About 3% of SSc patients have autoantibodies against the catalytic subunit of telomerase, hTERT [92]. In the shelterin complex (a group of six proteins that bind telomeres), telomeric repeat-binding factor-1 is one of the more frequently identified autoantigens in this category and was found in 9% of patients [92]. The autoantibody against telomeric repeat-binding factor-1 was associated with shorter telomeres in patients with the autoantibody than in those without it. Furthermore, this observation was specific to SSc, as patients with other autoimmune diseases exhibited the same frequency as the controls. However, a similar frequency of the anti-telomere repeat binding factor-1 was observed in patients with idiopathic pulmonary fibrosis, suggesting that telomere-targeting antibodies might trigger fibrotic lung disease [92]. Other, less frequently observed autoantibodies were found to target telomeric repeat binding factor-2, protection of telomeres-1, adrenocortical dysplasia homolog, TERF1-interacting nuclear factor-2, and repressor/activator protein-1 [92]. Beyond the core telomerase and shelterin proteins, studies have also identified autoantibodies against other proteins associated with telomere biology and maintenance. These include THO complex, homeobox-containing-1, RuvBL1, and RuvBL2 [93,94,95]. A timeline of significant findings on telomere attrition and telomere-associated autoantibodies is depicted in Figure 3.

## 5. Telomere Attrition and Chromosomal Instability in SSc-like Fibrotic Diseases

The tendency for chromosomal breakage and rearrangements, as well as clastogenic factors, has been reported in other diseases and may, in fact, be a feature of inflammation rather than a disease-specific characteristic of SSc. Clastogenic factors have been found in Bloom syndrome [96], lupus [27,97], rheumatoid arthritis [27,98], psoriasis [99], and hepatitis C [100]. They are also found in the autoimmune-prone New Zealand black mouse [27].

SSc and SSc-like diseases have been associated with various occupational exposures, and these chemicals have also been linked to chromosomal abnormalities. Increased chromosomal abnormalities have been associated with cyclophosphamide in SSc and rheumatoid arthritis, suggesting that this chemical increases the risk of chromosomal instability [101]. Various environmental exposures have been linked to SSc and SSc-like illnesses, and these exposures are also associated with chromosomal abnormalities (Table 2). The majority of these associations between SSc and various chemicals, solvents, or environmental toxins are primarily based on case studies or small cohorts, and, to date, it remains unclear whether these observations will hold up to scrutiny in larger patient groups. Studying causative chemical exposures in SSc is difficult, as we are all exposed to various chemicals in daily life. Whether this contributes to disease in certain individuals with specific genetic backgrounds will require intensive study.

The most well-known chemical that induces SSc-like disease in humans is bleomycin. Bleomycin is a glycopeptide antibiotic used as a chemotherapeutic agent. Bleomycin is now more extensively used to induce collagen synthesis in animal models of fibrotic diseases [138]. In humans, the drug-induced pathology is very similar, and they show skin hyperpigmentation, Raynaud’s phenomenon, and interstitial lung disease; however, these patients do not exhibit the classic autoantibodies that are diagnostic of SSc [139,140,141]. Bleomycin enters the cell and generates ROS, which breaks the DNA [142]. Paramagnetic resonance studies showed that bleomycin generates ROS, including superoxide, hydrogen peroxide, and the hydroxyl radical [143,144]. It was previously thought that ROS were primarily responsible for DNA damage; however, it is now accepted that bleomycin reacts with iron and oxygen to damage susceptible DNA sites [145]. The damage caused by bleomycin includes double-strand breaks, translocations, deletions, and the formation of dicentric chromosomes [102,103,104,105,106].

Vinyl chloride was one of the first chemicals associated with features mimicking SSc [107,108,109,110]. It can induce Raynaud’s phenomenon and scleroderma-like skin thickening in some individuals [111]. Vinyl chloride is metabolized in the liver into highly reactive intermediates, primarily chloroethylene oxide and chloroacetaldehyde, which can form DNA adducts that interfere with DNA replication and repair [146]. This leads to genetic mutations and large-scale chromosomal abnormalities in exposed individuals [112]. The types of chromosomal abnormalities observed in SSc, such as aneuploidy, breaks, acentric fragments, dicentric chromosomes, and sister chromatid exchange, are also observed with excessive exposure to vinyl chloride [112].

Other chemicals associated with SSc-like illnesses include trichloroethylene [113,114,115,116] and benzene [119,147,148], among other chemicals (Table 2). These chemicals also cause chromosomal instability upon exposure [122,149]. More recently, toxic oil syndrome [124,150] and nephrogenic fibrosing dermopathy have been associated with SSc-like diseases [151,152]. The cause of toxic oil syndrome in Spain in 1981 was the consumption of adulterated rapeseed oil intended for industrial use that was fraudulently sold as olive oil [153]. The industrial oil was contaminated with aniline, which is metabolized to phenylhydroxylamine and nitrosobenzene. These substances induce the production of ROS, which causes single- and double-strand DNA breaks [125]. Approximately 20,000 were exposed, and 1200 died [154]. Of those who survived, they developed a chronic neuromyopathy and scleroderma-like illness with similar features of collagen vascular disease [153] with overlapping forms of eosinophilic fasciitis [124]. In nephrogenic fibrosing dermopathy, gadolinium is the contrast agent that can bind non-covalently to DNA, causing indirect damage and inducing genotoxic effects, such as the formation of micronuclei and nuclear buds, likely through the generation of oxidative stress [155,156].

Several meta-analyses have reported associations between chemical exposures and the onset of SSc. Regarding occupational exposure to organic solvents, meta-analyses or systematic reviews reported increased relative risk and odds ratios with a greater risk in men [118,135,157,158,159]. SSc has been associated with geographical clustering in some areas, suggesting increased exposures to pollution or environmental toxins [160,161,162,163,164], although this observation did not always hold up to further analysis [165,166].

## 6. Inflammatory Triggers of Telomere Attrition and Chromosomal Instability in SSc and SSc-like Fibrotic Diseases

Currently, in SSc, and indeed in many other fibrotic diseases, we do not know what the trigger is that initiates fibrosis. However, once inflammation starts, it is a self-perpetuating mechanism. Inflammation and oxidative stress are core features that can cause telomere attrition and chromosomal damage. One fundamental activator of inflammation is the inflammasome [138] and other inflammatory pathways that drive fibrosis play a role [167]. Though not discussed here, the inflammasome is strongly correlated with SSc fibrosis [138], and can increase fibroblast turnover rates [168] and promote fibroblast differentiation [138]. The inflammasomes can be activated by oxidative stress [169] and discussed above SSc has elevated ROS. Furthermore, DNA damage can activate the inflammasome and cGAS-STING [22]. The other inflammatory pathways involved, such as AKT, G-protein-coupled receptors, MAP kinases, and WNT signaling, can promote cell proliferation [167]. To repair the damage, fibroblasts must divide more frequently to replace those lost or injured and this would contribute to telomere attrition. During inflammation, ROS are produced, and telomeres are highly susceptible to oxidative damage because the guanine base is readily oxidized [170]. The damage creates 8-oxo-deoxyguanosine lesions in the telomere [171,172], and if this modification escapes repair, it could lead to the misincorporation of an A within the telomere repeat [173]. Barnes et al. show that 8-oxo-deoxyguanosine disrupts telomere replication, increasing their fragility [174], and Fauquerel showed that this led to telomere loss [175]. NADPH oxidase-mediated superoxide production is strongly associated with shorter telomeres and atherosclerosis [176], implicating a correlation with telomere shortening in SSc [82] with its increased NADPH activity [31]. We believe that if the ROS production exceeds the available antioxidants, as seen in SSc [32], the cell’s ability to repair the sheer number of breaks is overwhelmed. This faulty repair could lead to deletions, translocations, and other chromosomal aberrations seen in SSc. Here, it is likely that oxidative stress from inflammation is the primary culprit. ROS are potent clastogens that cause both single-strand and double-strand breaks. Once telomeres become critically short, the cell’s machinery can no longer distinguish the natural chromosome end from a double-strand break, and the cell tries to fix this by fusing it to another chromosome. This faulty repair can create a dicentric chromosome, which undergoes a random break during cell division. This initiates a catastrophic breakage-fusion-bridge cycle that leads to massive and ongoing chromosomal instability [177]. This abnormality occurs when the telomere breaks or is lost, leading to the fusion of sister chromatids. During cell division, the fused structure forms a bridge between the separating poles of the cell, which then breaks unevenly [178]. This creates daughter cells with unstable, telomere-less chromosomes such as that seen in SSc [18], allowing the cycle to repeat.

We question whether these genomic alterations in SSc are a specific feature of the disease or a generalized phenomenon of fibrosis caused by increased inflammation and oxidative stress. Several other fibrotic disorders also feature chromosomal instability, telomere attrition, and loss of heterozygosity. Werner’s syndrome is caused by a mutation in the Werner gene, which is involved in DNA repair. There is a loss of heterozygosity that leads to genomic instability, accelerated cell senescence, and the characteristic skin-like fibrosis seen in SSc patients [179,180]. Ataxia-telangiectasia is caused by mutations in the ataxia-telangiectasia gene, another master regulator of DNA damage repair. While this is not a classic fibrotic disease, the loss of a functional gene contributes to oxidative stress and chronic inflammation, which can drive fibrosis in some patients [181,182,183]. In idiopathic pulmonary fibrosis, a significant subset of familial and even sporadic cases is caused by mutations in genes essential for telomere maintenance [184,185]. Even without mutations, accelerated telomere shortening is often observed in lung epithelial cells of patients, suggesting it is a key part of the disease process [185,186]. Dyskeratosis congenita is a rare, inherited bone marrow failure syndrome, a classic telomere biology disorder, and is caused by mutations in genes involved in telomere maintenance [187,188]. Patients often develop pulmonary fibrosis and liver cirrhosis as major complications, directly linking telomere dysfunction to fibrosis in these organs [187,188]. Aplastic anemia primarily affects the bone marrow, but some patients also develop telomere attrition, as well as pulmonary fibrosis or liver disease, further highlighting the systemic impact of telomere dysfunction [189]. Telomere shortening is also observed in hepatocytes during chronic liver injury and the progression to cirrhosis, regardless of the initial cause (e.g., viral hepatitis, alcohol, etc.). This shortening contributes to cell senescence and liver fibrosis [190]. Fanconi anemia is caused by mutations in genes involved in repairing DNA crosslinks. Fanconi anemia leads to significant chromosomal instability, and while it is primarily known for bone marrow failure and cancer risk, liver fibrosis can occur [191].

## 7. Discussion

The reviewed literature established a strong historical basis for the presence of significant genomic anomalies, specifically chromosomal instability and telomere attrition in the pathogenesis of SSc. We propose that these factors are linked in a self-perpetuating cycle that drives the chronic, fibrotic nature of the disease. One key finding on genomic anomalies in SSc includes chromosomal instability. SSc is marked by numerous chromosomal aberrations, including micronuclei, translocations, dicentric chromosomes, acentric fragments, and deletions, which are seen in lymphocytes, fibroblasts, and bone marrow. This instability is driven by clastogenic factors, such as inosine triphosphate and diphosphate, and by oxidative stress from heightened ROS production, which overwhelms the patient’s low antioxidant defenses. The presence of extensive genomic damage, including hypermutation (with a signature common in cancers, SBS93) and loss of heterozygosity, raises the question of whether this instability contributes to the increased incidence of cancer observed in SSc patients [41].

Research investigating clastogenic factors and ROS in SSc requires a modern mechanistic bridge to better understand their roles and define the relationship between the clastogenic factor (e.g., inosine triphosphate/diphosphate or other) and elevated NADPH oxidase activity. Understanding this mechanistic intersection might help to identify the initial biochemical trigger of chromosomal breakage. Inosine triphosphate and inosine diphosphate are central intermediates in the cell’s purine metabolism pathway. They are classified as noncanonical purine nucleotides and are not typically incorporated into DNA or RNA. Under normal conditions, they do not accumulate to high levels because inosine triphosphatase hydrolyzes them to inosine monophosphate and pyrophosphate [192]. This process prevents the incorporation of inosine into nucleic acids, which would cause DNA damage and mutagenesis [193]. The elevated ROS generated by NADPH oxidase causes irreversible damage to DNA, lipids, and enzymes. This extensive cellular damage and high metabolic turnover lead to the production of noncanonical purine nucleotides. The accumulation of inosine triphosphate/diphosphate would further drive genomic instability and inflammation, which, in turn, activates more NADPH oxidase, perpetuating the inflammatory cycle.

A second key finding is telomere attrition. The mechanism of telomere loss is primarily caused by chronic inflammation and oxidative stress, as the guanine-rich telomeres are highly susceptible to damage from ROS [194]. This damage disrupts their replication and increases telomere fragility. The critically short telomeres activate the DNA damage response, leading to faulty DNA repair (e.g., breakage–fusion–bridge cycles) and the chromosomal instability seen in the disease. While autoantibodies targeting shelterin components (e.g., TRF1) and telomerase (hTERT) have been identified in a subset of patients, their functional consequences remain to be investigated. Currently, we do not know whether they directly block telomerase activity or shelterin binding, thereby accelerating telomere loss. Further investigation of telomerase activity in various SSc cell types (lymphocytes, fibroblasts, granulocytes, etc.) is required to determine whether reduced activity is generalized or lineage-specific, and whether this relates to telomerase genotype frequencies [91].

Incorporating recent observations on the cyclic GMP–AMP synthase (cGAS)-stimulator of interferon genes (STING) pathway, damage-associated molecular patterns (inflammasome activation), and the senescence-associated secretory phenotype would provide a clearer understanding of how the core findings of genomic instability and telomere attrition directly lead to the chronic inflammation and tissue fibrosis seen in SSc. This integration could bridge the gap between DNA damage (possibly the cause) and inflammation/fibrosis (the effect) by defining the specific signaling pathways that sense genomic stress. cGAS-STING is one critical molecular sensor that directly links genomic instability to inflammatory and immune responses. When the cell experiences significant genomic stress, fragments of damaged DNA or DNA leakage from damaged mitochondria can accumulate in the cytosol. cGAS senses this aberrant cytosolic DNA. Upon binding the DNA, cGAS produces the secondary messenger cGAMP, which then activates the adaptor protein STING. STING activation triggers the production of type I interferons and other pro-inflammatory cytokines, driving chronic inflammation. This chronic inflammation is a well-established driver of enhanced fibroblast proliferation and differentiation, ultimately leading to fibrosis. Activation of cGAS-STING has been associated with other fibrotic diseases [195,196,197,198,199] and this pathway has garnered much attention in the last 5 years. Indeed, inhibitors of cGAS or STING have been identified that ameliorate fibrosis [200,201].

Damage-associated molecular patterns are endogenous “danger signals” released by damaged or dying cells. Genomic instability directly creates these signals that can activate the inflammasomes, and this would occur in parallel with cGAS-STING. These danger signals are released during inflammation-induced cell death (apoptosis/necrosis) or when telomere attrition leads to micronuclei formation. The damaged DNA fragments function as the “on switch” for inflammation. They are known to activate the inflammasome, which is strongly correlated with fibrosis [138]. This activation releases potent pro-fibrotic cytokines, locking the tissue into an inflammatory state. Critically short telomeres, DNA damage, and chronic stress often force cells, especially fibroblasts, into a state of cellular senescence. Senescent cells, rather than lying dormant, become highly secretory. This potent secretome is characterized by the release of numerous inflammatory and profibrotic molecules, including pro-inflammatory cytokines (e.g., IL-6, IL-1α, and IL-1β), chemokines, and matrix remodeling enzymes. The secretome directly fuels the fibrotic cycle by driving further inflammation, recruiting immune cells, and altering the tissue microenvironment to favor collagen deposition and fibroblast differentiation. This ensures the vicious cycle of inflammation, genomic damage, and fibrosis is perpetuated (Figure 4).

Overall, we propose a model of inflammation, ROS production, telomere attrition, and chromosomal instability that initiates a feedback loop, leading to further inflammation and DNA damage, which, in turn, increases telomere attrition and chromosomal instability. Then, the chromosomal instability and damaged DNA act as potent danger signals that further activate inflammatory pathways, such as the inflammasome [138], cGAS-STING, and other inflammatory pathways involved in fibrosis [167]. This can also trigger cellular senescence, leading cells to secrete more pro-inflammatory cytokines, which, in turn, generate even more inflammation and oxidative stress. In essence, these pathways explain that genomic damage is not just an effect but can also be a catalyst, translating the chromosomal damage signal into a chronic, self-perpetuating inflammatory and fibrotic program.

## Figures and Tables

**Figure 1 genes-16-01466-f001:**
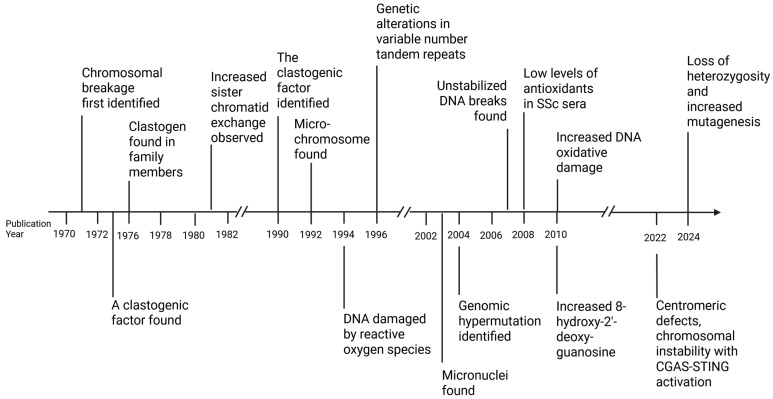
Historical observations of chromosomal instability in SSc. Chromosomal instability and various genetic alterations have been studied sporadically for the last 50+ years. Created in BioRender. Artlett, C. (2025) https://BioRender.com/lbjflxo. Accessed 19 November 2025.

**Figure 2 genes-16-01466-f002:**
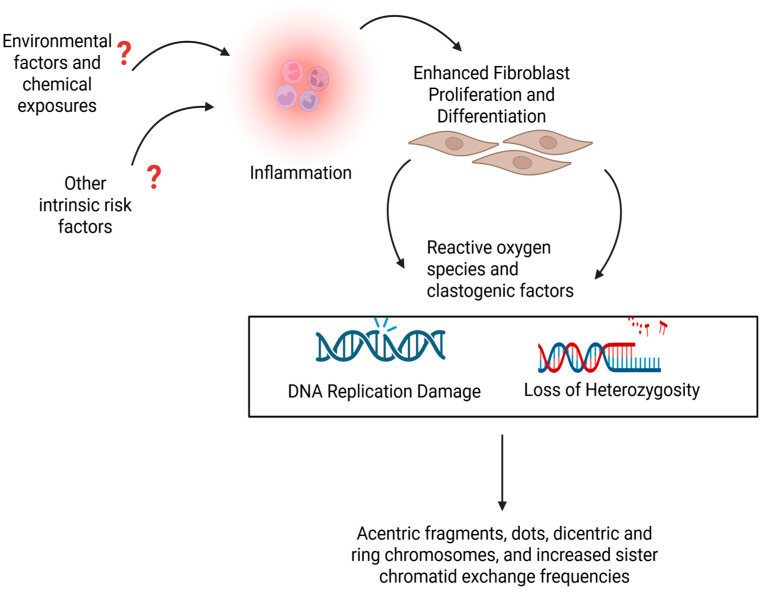
Hypothetical mechanism for the chromosomal instability seen in SSc. Created in BioRender. Artlett, C. (2025) https://BioRender.com/f41g136. Accessed 19 November 2025.

**Figure 3 genes-16-01466-f003:**
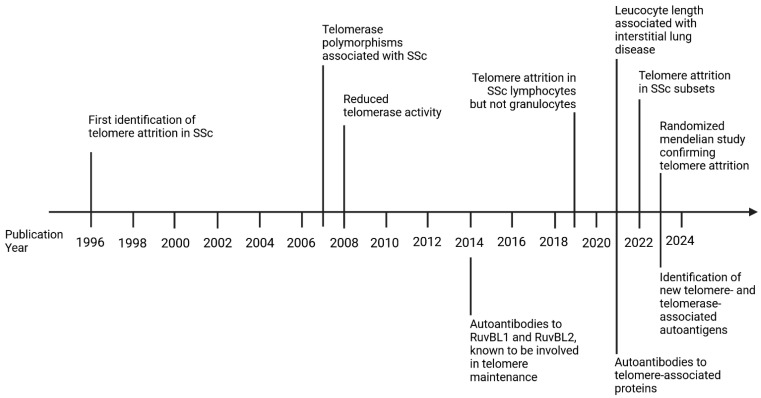
Timeline for the identification of telomere attrition and telomerase-associated autoantigens. Telomere attrition in SSc has received little attention, but in recent years, there has been increased interest in understanding why telomeric sequences are lost at a greater rate in SSc. Created in BioRender. Artlett, C. (2025) https://BioRender.com/il4sfod. Accessed 19 November 2025.

**Figure 4 genes-16-01466-f004:**
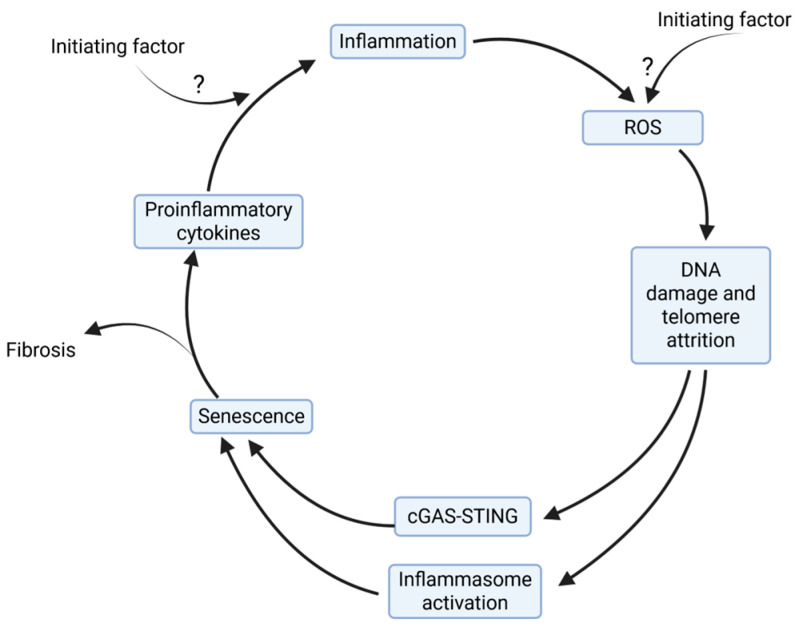
A proposed model for the chronic, self-perpetuating nature of inflammation, telomere loss, and chromosomal instability. The initiating stimulus, whether an environmental toxin or pathogenic, promotes inflammation, leading to the production of ROS. This, in turn, damages DNA, causing chromosomal instability and telomere attrition. The damaged DNA causes parallel activation of cGAS-STING and inflammasomes, leading to fibroblast senescence. Senescent fibroblasts secrete proinflammatory cytokines and fibrotic mediators, thereby increasing extracellular matrix deposition, a feature of fibrosis. The proinflammatory cytokines also exacerbate inflammation, leading to increased ROS production and perpetuating the cycle. The initiating factor’s entry into the cycle is unknown, but it could directly trigger inflammation or ROS. Created in BioRender. Artlett, C. (2025) https://BioRender.com/pswefnm. Accessed 21 November 2025.

**Table 2 genes-16-01466-t002:** Chemicals Associated with SSc-like illnesses and Chromosomal Instability *.

Chemical	SSc-like Manifestations	Chromosomal Abnormalities	References
Bleomycin	Skin and lung fibrosis	Double-strand breaks, translocations, deletions, and dicentric chromosomes	[102,103,104,105,106]
Vinyl chloride	SSc-like skin thickening, liver fibrosis, lung fibrosis, Raynaud’s phenomenon	Not directly responsible for causing DNA damage, but its metabolites can induce DNA damage, causing breaks, fragments, deletions, dicentric chromosomes, and translocations	[107,108,109,110,111,112]
Trichloroethylene	SSc-like skin changes, Raynaud’s phenomenon, lung and kidney fibrosis	Increased sister chromatid exchange, increased DNA methylation, and aneuploidy	[113,114,115,116,117,118]
Benzene	SSc-like skin changes	Aneuploidy, increased sister chromatid exchange, micronuclei, hypermethylation, and other chromosomal aberrations	[117,119,120,121]
Xylene	Sclerodermatous skin changes	Breaks, aneuploidy, hypermethylation	[119,121,122,123]
Toxic oil (aniline)	SSc-like skin changes, Raynaud’s phenomenon, pulmonary hypertension, collagen vascular disease	Single- and double-strand breaks, and sister chromatid exchange	[124,125,126]
Cocaine	DcSSc, digital ulcers, and SSc renal crisis	Genomic instability and aneuploidy	[127,128,129]
Silica	Raynaud’s phenomenon, skin thickening,	Micronuclei and DNA breaks, and other chromosomal aberrations	[118,130,131,132,133,134,135]
L-tryptophan (Peak E **)	Dermal fibrosis and eosinophilic fasciitis	Indirectly causes chronic inflammation and oxidative stress that can lead to chromosomal instability	[136,137]

* Most of these associations are based on case studies or small cohorts. However, several publications have employed meta-analyses, thereby strengthening the correlation between the two. These meta-analyses are from [118,120,135]. Whether chromosomal abnormalities present first and then drive SSc is currently unknown. ** Originally known as Peak E, it was later identified to be 1,1′-ethylidenebis (tryptophan).

## Data Availability

No new data were created or analyzed in this study.

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
