# Peer review of "Chromosomal Instability and Telomere Attrition in Systemic Sclerosis: A Historical Perspective"

_genes, 2025, doi:10.3390/genes16121466_

Round 1
Reviewer 1 Report
Comments and Suggestions for Authors
This comprehensive review offers a great review of both historical and recent advances in our understanding of chromosomal instability and telomere attrition in SSc. The author effectively places genomic anomalies, such as chromosomal alterations and premature telomere erosion, at the center of disease progression, highlighting their dynamic contribution to SSc pathogenesis. The manuscript is well written, and the referencing is thorough and balanced. The reviewer has only a few suggestions:
(1) The section addressing autoantibodies is particularly intriguing. To further enhance its clarity and impact, the reviewer recommends adding a summary table that organizes the key findings and implications associated with each autoantibody described in the text.
(2) While the author provides an excellent overview of DNA instability, the manuscript could be strengthened by a more detailed discussion on potential mechanisms linking chromatin instability to fibrosis. Incorporating recent advances, such as the roles of the cGAS-STING pathway, damage-associated molecular patterns, and the senescence-associated secretory phenotype, would give readers a clearer understanding of how these processes may bridge genomic instability and tissue fibrosis, an area of active research in the field.
Author Response
This comprehensive review offers a great review of both historical and recent advances in our understanding of chromosomal instability and telomere attrition in SSc. The author effectively places genomic anomalies, such as chromosomal alterations and premature telomere erosion, at the center of disease progression, highlighting their dynamic contribution to SSc pathogenesis. The manuscript is well written, and the referencing is thorough and balanced. The reviewer has only a few suggestions:
Response: thank you so much for the supportive comments of my manuscript. I have addressed the following issues that you have raised.
Comment 1: The section addressing autoantibodies is particularly intriguing. To further enhance its clarity and impact, the reviewer recommends adding a summary table that organizes the key findings and implications associated with each autoantibody described in the text.
Response 1: I have included a new table (Table 1) in the text that summarizes key findings and potential implications
Comment 2: While the author provides an excellent overview of DNA instability, the manuscript could be strengthened by a more detailed discussion on potential mechanisms linking chromatin instability to fibrosis. Incorporating recent advances, such as the roles of the cGAS-STING pathway, damage-associated molecular patterns, and the senescence-associated secretory phenotype, would give readers a clearer understanding of how these processes may bridge genomic instability and tissue fibrosis, an area of active research in the field.
Response: The discussion has been completely overhauled, incorporating the cGAS-STING pathway, DAMPs, and the senescence-associated secretory phenotype.
Reviewer 2 Report
Comments and Suggestions for Authors
TITLE: Chromosomal Instability and Telomere Attrition in Systemic Sclerosis: A Historical Perspective by Carol M. Artlett.
SUMMARY: The author reviews evidence of chromosomal instability and telomere shortening in systemic sclerosis and discusses potential links to autoantibodies, inflammation, reactive oxygen species, and fibrosis. A model is proposed in which inflammation drives reactive oxygen species production, causing chromosomal damage, leading in turn to more inflammation, with the implication that this cycle contributes to fibrosis.
OVERALL COMMENTS:
Sections 2 and 4 are insightful reviews of empiric evidence demonstrating chromosomal abnormalities and telomere shortening in SSc, including references to early SSc literature that does not receive much attention in most reviews on SSc pathogenesis. The rest of the review is more speculative, pulling from a broad assortment of observations to generate a thesis as outlined in Figure 3. While it is appropriate to hypothesize in a review article, it should be acknowledged that the proposed model has not been rigorously demonstrated to drive SSc pathogenesis. The review might be improved with an associated discussion on how this model could be empirically tested in future research and clinical trials. I am also concerned that some of the cited literature is over-interpreted, with the potential for a misleading effect.
Specific critiques as follows:
-The statement in line 67 “SSc does not have a genetic inheritance pattern” is oversimplified and seems dismissive of the large body of evidence that several heritable genetic loci contribute to SSc susceptibility.
-In lines 105-107 referencing an open-label trial of a topical superoxide dismutase intervention (reference 34), the reference should be contextualized by noting that the intervention was in a small number of subjects with an unclear outcome measure and that this has not been validated in subsequent studies and not in a randomized placebo-controlled trial.
-In line 272, the statement “Neutrophils play a prominent role in SSc…” may be an overinterpretation of available data. The cited paper (reference 80) showed only an association with ILD severity but not a causal relationship. One could argue since SSc is generally not responsive to glucocorticoid treatment like most neutrophilic diseases are, that neutrophils do not play a prominent role. Not that this needs to be discussed in the article, but I suggest toning down the current statement to better reflect the findings of this study.
-In the timeline (Figure 2), there is mention of telomerase polymorphisms associated with SSc, but I did not find mention of this in the text.
-In lines 333-335, it is unclear what is meant to be implied by the statement about cyclophosphamide in SSc. Cyclophosphamide was in fact the first intervention ever shown in a placebo-controlled randomized trial to be beneficial for treatment of SSc-associated ILD (Scleroderma Lung Study, Tashkin et al., NEJM. 2006).
-I suggest more caution in how Table 1 is presented and contextualized. An unfamiliar reader could easily interpret this section to mean that each of the listed chemicals has been robustly linked to SSc. I suggest moving the footnote (about most of the chemicals being based on case studies) to the main text to contextualize the discussion, and perhaps elaborate on why that is such a limitation (potential for publication bias, lack of robust epidemiologic data demonstrating statistical association). I find inclusion of cocaine being associated with DcSSc to be particularly questionable. Regarding the footnote about several of the publications using meta-analyses, it was hard to tell which ones, so I recommend citing the specific studies that are being referenced in this statement.
-It seems a bit selective to state that “The initiating hit” in SSc inflammation is the inflammasome, without referencing other inflammatory pathways implicated in SSc pathogenesis. Consider citing a review on this subject.
Minor critiques:
-In line 45 discussing anticentromere antibody in lcSSc, I suggest replacing “primary diagnostic autoantibody” with “most common autoantibody” as the auto-antibodies in SSc are not diagnostic but are included in the 2013 classification (not diagnostic) criteria.
-In lines 47-48 in reference to Topo-I and RNA Pol III antibodies in dcSSc, I suggest replacing “characterized by” with “associated with.”
-In line 186 it should be clarified that the antigen recognized by the Scl-70 antibody (not the autoantibody itself) has a molecular weight of approximately 70 kDa. I also recommend adding the citation in which the target of the Scl-70 antibody was identified as Topoisomerase I (Shero et al., Science 1986;231:737-40).
-Regarding the prevalence of RNA Polymerase III antibody in SSc (line 212), it varies by ethnicity and geographic location. Consider substituting the current citation with this recent review—Elhannani et al., Rheumatology (Oxford). 2025 Jul 17:keaf392.
-In the section on telomere shortening in SSc, consider adding a citation of this recent article—Yang et al., Arthritis Rheumatol. 2025 Sep 18.
-the first paragraph of section 6 might be better framed as a separate section on inflammatory triggers of chromosomal damage before transitioning to the Discussion section.
Author Response
Comment: The statement in line 67 “SSc does not have a genetic inheritance pattern” is oversimplified and seems dismissive of the large body of evidence that several heritable genetic loci contribute to SSc susceptibility.
Response: You are right that there are regions in the genome that are associated with SSc, but what I meant was a familial inheritance pattern. I have clarified this statement.
Comment: In lines 105-107 referencing an open-label trial of a topical superoxide dismutase intervention (reference 34), the reference should be contextualized by noting that the intervention was in a small number of subjects with an unclear outcome measure and that this has not been validated in subsequent studies and not in a randomized placebo-controlled trial.
Response: Thank you. This is a valid point, and I have added your concerns for this study and discussed the lack of follow on studies. This now starts line 115.
Comment: In line 272, the statement “Neutrophils play a prominent role in SSc…” may be an overinterpretation of available data. The cited paper (reference 80) showed only an association with ILD severity but not a causal relationship. One could argue since SSc is generally not responsive to glucocorticoid treatment like most neutrophilic diseases are, that neutrophils do not play a prominent role. Not that this needs to be discussed in the article, but I suggest toning down the current statement to better reflect the findings of this study.
Response: I changed the sentence to say “Neutrophils may play a role in SSc……..” on line 295.
Comment: In the timeline (Figure 2), there is mention of telomerase polymorphisms associated with SSc, but I did not find mention of this in the text.
Response: it had been included in the original text on line 319.
Comment: In lines 333-335, it is unclear what is meant to be implied by the statement about cyclophosphamide in SSc. Cyclophosphamide was in fact the first intervention ever shown in a placebo-controlled randomized trial to be beneficial for treatment of SSc-associated ILD (Scleroderma Lung Study, Tashkin et al., NEJM. 2006).
Response: I am not denying that cyclophosphamide can be beneficial for SSc. All I am stating is that the study reports an increase in chromosomal anomalies in SSc and RA treated with cyclophosphamide.
Comment: I suggest more caution in how Table 1 is presented and contextualized. An unfamiliar reader could easily interpret this section to mean that each of the listed chemicals has been robustly linked to SSc. I suggest moving the footnote (about most of the chemicals being based on case studies) to the main text to contextualize the discussion, and perhaps elaborate on why that is such a limitation (potential for publication bias, lack of robust epidemiologic data demonstrating statistical association). I find inclusion of cocaine being associated with DcSSc to be particularly questionable. Regarding the footnote about several of the publications using meta-analyses, it was hard to tell which ones, so I recommend citing the specific studies that are being referenced in this statement.
Response: I have moved my footnote to the text. I appreciate your questioning about the inclusion of cocaine, but I am just reporting possible associations that I found in the literature. I have also noted in the footnote those references that were meta-analyses.
Comment: It seems a bit selective to state that “The initiating hit” in SSc inflammation is the inflammasome, without referencing other inflammatory pathways implicated in SSc pathogenesis. Consider citing a review on this subject.
Response: I have included a review by Sadatpour (2025) on the subject.
Minor critiques:
Comment: In line 45 discussing anticentromere antibody in lcSSc, I suggest replacing “primary diagnostic autoantibody” with “most common autoantibody” as the auto-antibodies in SSc are not diagnostic but are included in the 2013 classification (not diagnostic) criteria.
Response: I have changed this wording
Comment: In lines 47-48 in reference to Topo-I and RNA Pol III antibodies in dcSSc, I suggest replacing “characterized by” with “associated with.”
Response: I have made that change.
Comment: In line 186 it should be clarified that the antigen recognized by the Scl-70 antibody (not the autoantibody itself) has a molecular weight of approximately 70 kDa. I also recommend adding the citation in which the target of the Scl-70 antibody was identified as Topoisomerase I (Shero et al., Science 1986;231:737-40).
Response: Agreed that my wording was incorrect. I have made these changes and included the Shero reference, now on line 201.
Comment: Regarding the prevalence of RNA Polymerase III antibody in SSc (line 212), it varies by ethnicity and geographic location. Consider substituting the current citation with this recent review—Elhannani et al., Rheumatology (Oxford). 2025 Jul 17:keaf392.
Response: I have included this reference, line 233.
Comment: In the section on telomere shortening in SSc, consider adding a citation of this recent article—Yang et al., Arthritis Rheumatol. 2025 Sep 18.
Response: I have added this reference on line 301, reference 86
Comment: The first paragraph of section 6 might be better framed as a separate section on inflammatory triggers of chromosomal damage before transitioning to the Discussion section.
Response: I have made this a separate section, called 6. Inflammatory Triggers of Telomere Attrition and Chromosomal Instability in SSc and SSc-like Fibrotic Diseases. The discussion has been completely changed.
Round 2
Reviewer 2 Report
Comments and Suggestions for Authors
The author has addressed the concerns of the reviewers.